# GSKIP-Mediated Anchoring Increases Phosphorylation of Tau by PKA but Not by GSK3beta via cAMP/PKA/GSKIP/GSK3/Tau Axis Signaling in Cerebrospinal Fluid and iPS Cells in Alzheimer Disease

**DOI:** 10.3390/jcm8101751

**Published:** 2019-10-21

**Authors:** Huey-Jiun Ko, Shean-Jaw Chiou, Yu-Hui Wong, Yin-Hsuan Wang, Yun-Ling Lai, Chia-Hua Chou, Chihuei Wang, Joon-Khim Loh, Ann-Shung Lieu, Jiin-Tsuey Cheng, Yu-Te Lin, Pei-Jung Lu, Ming-Ji Fann, Chi-Ying Huang, Yi-Ren Hong

**Affiliations:** 1Graduate Institute of Medicine, College of Medicine, Kaohsiung Medical University, Kaohsiung 807, Taiwan; o870391@yahoo.com.tw (H.-J.K.); sheanjaw@kmu.edu.tw (S.-J.C.); kime036@gmail.com (Y.-H.W.); 4a1h0010@gmail.com (Y.-L.L.); lucifer0408@hotmail.com (C.-H.C.); jokhlo@kmu.edu.tw (J.-K.L.); e791125@gmail.com (A.-S.L.); 2Department of Biochemistry, Kaohsiung Medical University, Kaohsiung 807, Taiwan; 3Department of Medical Research, Kaohsiung Medical University Hospital, Kaohsiung 807, Taiwan; 4Brain Research Center, National Yang-Ming University, Taipei 11221, Taiwan; yuhui.wong@gmail.com; 5Department of Biotechnology, Kaohsiung Medical University, Kaohsiung 807, Taiwan; chwang@kmu.edu.tw; 6Department of Neurosurgery, Kaohsiung Medical University Hospital, Kaohsiung 807, Taiwan; 7Department of Biological Sciences, National Sun Yat-Sen University, Kaohsiung 804, Taiwan; tusya@mail.nsysu.edu.tw; 8Section of Neurology, Kaohsiung Veterans General Hospital, Kaohsiung 813, Taiwan; ytlin@vghks.gov.tw; 9Institute of Clinical Medicine, School of Medicine, National Cheng Kung University, Tainan 701, Taiwan; pjlu2190@mail.ncku.edu.tw; 10Department of Life Sciences and Institute of Genome Sciences and Brain Research Center, National Yang-Ming University, Taipei 11221, Taiwan; mjfann@ym.edu.tw; 11Institute of Biopharmaceutical Sciences, National Yang-Ming University, Taipei, Taiwan; Department of Biotechnology and Laboratory Science in Medicine, National Yang-Ming University, Taipei 112, Taiwan

**Keywords:** PKA/GSKIP/GSK3β/Tau axis, SH-SY5Y, iPS cells, cerebrospinal fluid, Alzheimer’s disease

## Abstract

Based on the protein kinase A (PKA)/GSK3β interaction protein (GSKIP)/glycogen synthase kinase 3β (GSK3β) axis, we hypothesized that these might play a role in Tau phosphorylation. Here, we report that the phosphorylation of Tau Ser409 in SHSY5Y cells was increased by overexpression of GSKIP WT more than by PKA- and GSK3β-binding defective mutants (V41/L45 and L130, respectively). We conducted in vitro assays of various kinase combinations to show that a combination of GSK3β with PKA but not Ca^2+^/calmodulin-dependent protein kinase II (CaMK II) might provide a conformational shelter to harbor Tau Ser409. Cerebrospinal fluid (CSF) was evaluated to extend the clinical significance of Tau phosphorylation status in Alzheimer’s disease (AD), neurological disorders (NAD), and mild cognitive impairment (MCI). We found higher levels of different PKA–Tau phosphorylation sites (Ser214, Ser262, and Ser409) in AD than in NAD, MCI, and normal groups. Moreover, we used the CRISPR/Cas9 system to produce amyloid precursor protein (*APP*^WT/D678H^) isogenic mutants. These results demonstrated an enhanced level of phosphorylation by PKA but not by the control. This study is the first to demonstrate a transient increase in phosphor-Tau caused by PKA, but not GSK3β, in the CSF and induced pluripotent stem cells (iPSCs) of AD, implying that both GSKIP and GSK3β function as anchoring proteins to strengthen the cAMP/PKA/Tau axis signaling during AD pathogenesis.

## 1. Introduction

GSK3β interaction protein (GSKIP) is the smallest A-kinase anchor protein (AKAP) [1,2]. It functions as a cytosolic scaffolding protein that can bind protein kinase A (PKA) and glycogen synthase kinase 3β (GSK3β). GSKIP retains the PKA RII binding sites at residue V41/L45 and the GSK3β binding domain at residue L130 [1,2,3,4,5,6]. The latter is known to act synergistically with cAMP/PKA signaling to inhibit GSK3β activity [1,7]. GSKIP that directly interacts with GSK3β and negatively regulates GSK3β signaling is required for negative Wnt signaling regulation via a cytoplasmic destruction complex that targets β-catenin for degradation [5,6,8,9]. GSKIP can compete for GSK3β binding, resulting in control of the β-catenin stabilizing phosphorylation at Ser675 by PKA [9]. Its interaction with GSK3β facilitates control of the phosphorylation at Ser-33/37/Thr-41, which destabilizes β-catenin. The effect of GSKIP on β-catenin may be as a scavenger because it recruits GSK3β away from the destruction complex without forming a complex with β-catenin [8]. According to our previous findings, GSK3β binding of GSKIP is involved in the control of neurite outgrowths in the neuronal-like SH-SY5Y cell line, suggesting a role in neural development [5]. Furthermore, because of the dual binding sites of GSKIP, we previously proposed that GSKIP and GSK3β form a complex and likely act as anchoring proteins in the cAMP/PKA/Drp1 signaling axis to modulate Drp1 phosphorylation, which may provide neuroprotection against H_2_O_2_-induced oxidative stress in the neuron-like SH-SY5Y cell line [7]. Loss of GSKIP expression results in embryonic lethality attributable to a developmental defect in palatal shelf fusion, as determined through knockout animal models [10]. GSKIP has been characterized as one of the predisposing genes of familial myeloproliferative neoplasms in gene overexpression studies [11,12]. More recently, the loss of the peroxisome proliferator-activated receptor gamma-E2F1 (PPARγ-E2F1) axis in the lungs was found to lead to a deficient complex with Wnt signaling that caused downregulation of GSKIP and eventually resulted in decreased angiogenesis, indicating that GSKIP may be involved in PPARγ-related angiogenic potential in mature pulmonary microvascular endothelial cells (PMVECs) through E2F1 [13]. Collectively, these findings prompt us to believe that GSKIP has a role in neurodevelopment. However, the PKA/GSKIP/GSK3β axis and the PKA and GSK3 signaling pathways involved in the pathogenesis of neurodegenerative disease events remain unclear. Clarifying the roles of these would help advance drug development for neuron-related disorders such as Alzheimer’s disease (AD).

AD is a neurodegenerative disorder and the main form of dementia, accounting for up to 60–70% of dementia cases; its prevalence is highly associated with advanced age [14]. The worldwide prevalence of AD was 33 million patients in 2018. By 2050, an estimated 107 million individuals worldwide will be living with the disease [15]. The characteristic neuropathological changes that are found in patients with AD are the presence of extracellular amyloid-β (Aβ) deposition and the formation of intracellular neurofibrillary tangles (NFTs) [16]. Tau is a normal, unfolded, highly soluble protein that plays a critical role in tubulin assembly and stabilization of microtubules, thereby promoting normal function and axonal localization of neurons, and its hyperphosphorylation plays a key role in the pathogenesis of AD; Tau undergoes conformational changes in which the conversion of Tau monomer to Tau oligomer induces the aggregation of Tau into a paired helical filament, leading to the formation of NFTs in AD [17]. Several studies have focused on the Aβ-targeting pathology as a potential treatment for AD; however, clinical evidence has shown that antiamyloid treatment for AD has not been effective at slowing the progression of the disease. Because tauopathies are indicated in dementia and AD, attention has shifted toward researching the pathology of Tau proteins and the mechanisms involved as well as potential Tau-targeted therapies for AD. More recently, 3D pharmacophore drug discovery methods and molecular docking have been applied to discover potent phosphor-Tau inhibitors in AD [18].

Anomalous signal transduction is related to the causes of neurodegenerative diseases. According to in vitro and in vivo evidence concerning neurodegenerative diseases such as AD, GSK3β has a crucial role in memory impairment [19]. AD has been highly associated with GSK3β hyperactivity in advanced studies [19,20,21]. Notably, GSK3β phosphorylates Tau to reduce microtubule binding ability, and this results in microtubule destabilization that further leads to accumulation of Aβ peptides [21,22]. Another cause of the accumulation of Aβ and β-amyloid plaques is anomalous amyloid precursor protein (*APP*) metabolism that also leads to AD. There are multiple structures of Aβ peptide; one is Aβ 40, which exists mainly in cerebrospinal fluid (CSF), and another is Aβ 42, which is associated with the formation of β-amyloid plaques [16,23]. Aβ formation promotes the hyperactivity of GSK3β in abnormal-*APP*-metabolism induced pluripotent stem cells (iPSCs) and in animal models [19,24,25,26]. In recent years, we have focused on the effects of the regulatory mechanism of GSK3β and its binding proteins on neuron degeneration disease, especially GSKIP, which suppresses GSK3β activity and Tau phosphorylation [5,6].

We have revealed that PKA enzyme activity is required for the formation of a complex comprising PKA/GSKIP/GSK3β that influences the phosphorylation of dynamin related protein 1 (Drp1) at Ser637 and β-catenin at Ser675 [7,9]. Recently, Ser409 phosphorylation of Tau—a known PKA/GSK3β substrate—was revealed to be associated with the termination of neurons in AD [27]. In the present study, we sought to use the PKA/GSKIP/GSK3β axis as a platform on which to search for a novel pathway that regulates through both PKA and GSK3β and then use it to investigate the mechanism underlying neuron protection or degeneration. In addition, we studied the spatiotemporal characteristics of Tau phosphorylation via the cAMP/PKA/Tau axis complex to determine the mechanism of Tau in AD. Our data reveal that both GSKIP and GSK3β function as anchoring proteins that are involved in cAMP/GSKIP/GSK3β/PKA/Tau axis signaling and may play key roles in the development of AD. In particular, data suggest the feasibility of determining a standard criterion whereby AD can be detected by examining total Tau alone with PKA phosphor-Tau sites (Ser214, Ser262, and Ser409) in CSF and the transient increase in phosphor-Tau by PKA but not GSK3β in CSF or iPSCs. Phosphor-Tau anchored by GSKIP via PKA sites could act as novel diagnostic and therapeutic means to protect against different types of AD in personalized medicine.

## 2. Materials and Methods

### 2.1. Cell Culture

The human neuroblastoma SH-SY5Y and human renal epithelial (HEK293) cell lines (American Type Culture Collection, Manassas, VA, USA) were used for these experiments.

SH-SY5Y cells were cultured in Dulbecco’s modified Eagle’s medium (DMEM)/F12, and HEK293 cells were cultured in DMEM (GIBCO BRL Life Technologies, Invitrogen Life Technologies, Carlsbad, CA, USA) containing 10% fetal bovine serum, penicillin (100 U/mL), and streptomycin (100 μg/mL; Invitrogen Life Technologies, Carlsbad, CA, USA). Cells were maintained in 5% CO_2_ at 37 °C and were passaged every 72 h.

### 2.2. Transfection and RNA Interference

For transient transfections, SH-SY5Y and HEK293 cells were seeded onto glass coverslips at a density of 1 × 10^5^ cells using a 6-well plate 24 h prior to transfection. The following were transfected into the cells using Lipofectamine 3000 (Invitrogen Life Technologies, Carlsbad, CA, USA) according to the manufacturer’s instructions: pEGFP-C1, pEGFP-C1-GSKIP, pEGFP-C1-GSKIP (L130P), pEGFP-C1-GSKIP (V41/L45P) pEGFP-C1-Tau, pET32a-HA, pET32a-HA-GSKIP, pET32a-HA-GSKIP (L130P), pET32a-HA-GSKIP (V41/L45P), pET32a-HA-GSK3β, pET32a-HA-GSK3β (K85R), and pET32a-HA-GSK3β (K85M) plasmid DNA (2 μg). After a 24 h transfection, the cells were cultured in fresh medium and further assayed. Three of GSK3β siRNA and scrambled siRNA duplexes were used for RNA interference assays. For the siRNA-mediated knockdown of GSKIP expression, approximately 1 × 10^5^ or 1 × 10^6^ cells were plated onto 12-well plates or 100 mm dishes and left to grow overnight. The following day, cells were transfected with the siRNA duplex (final concentration, 50 nM) using Lipofectamine 3000 (Invitrogen Life Technologies, Carlsbad, CA, USA).

### 2.3. Cloning and DNA Sequencing

To construct pET32a-HA-GSK3β, pET32a-HA-GSKIP and pEGFP-C1-GSKIP plasmids were constructed separately by inserting PCR fragments into commercial HA or EGFP expression vectors.

GSKIP L130P (Leu 130 to Pro), V41/L45P (Val 41 and Leu 45 to Pro), GSK3β K85R (Lys 85 to Arg), and K85M (Lys 85 to Met) mutants were created using the site-directed mutagenesis technique with a QuikChange Lightning Site-Directed Mutagenesis kit (GE Healthcare, Sunnyvale, CA, USA). All experimental procedures were carried out following the manufacturer’s protocol, and mutated nucleotides were verified by DNA sequencing with an ABI PRISM 3730 Genetic Analyzer (Applied Biosystems, Forster City, CA, USA).

### 2.4. In Vitro Kinase Assay

In vitro kinase reactions were performed by mixing 1 μg of the recombinant active GSK3β, PKA, and Ca^2+^/calmodulin-dependent protein kinase II (CaMK II) (New England Biolabs, BioLabs, Ipswich, MA, USA), all of which were incubated at room temperature for 30 min under a reaction condition that entailed 20 μL of the kinase activity buffer (50 mM Tris HCl, 10 mM MgCl_2_, 1 mM dithiothreitol, 10 μM ATP, pH 7.5); the reaction was subsequently stopped by adding 10 μL of SDS-PAGE loading buffer. After denaturation at 95 °C for 1 min, protein phosphorylation was analyzed by immunoblotting with the indicated antibodies.

### 2.5. Coimmunoprecipitation

Coimmunoprecipitation (co-IP) assays were performed in the whole cell lysates of HEK293 cells. For co-IP, 500 μL of phosphate buffered saline (PBS) with protease and phosphatase inhibitors (1:100, Sigma-Aldrich, St. Louis, MO, USA) was used to dilute 500 μg cell extracts in lysis buffer (150 mM NaCl, 50 mM Tris HCl, pH 7.6, 1% NP-40, 0.1% SDS) overnight at 4 °C. Samples precleared through incubation with Protein A/G agarose beads (Oncogene Science) were added to the lysate, and the mixture was incubated with shaking for 2 h at 4 °C. The beads were spun down and washed three times with immune antibody precipitation assay buffer. Proteins binding to the beads were eluted by adding 20 μL of 2× electrophoresis sample buffer and analyzed through immunoblotting with an anti-HA (Cat#11666851001) or anti-GFP (Cat#11814460001) antibody (Roche, Basel, Switzerland).

### 2.6. Study Groups and CSF Samples

Samples from a total of 20 participants were collected and categorized into four groups: Normal, AD, neurological disorders (NAD), and mild cognitive impairment (MCI). CSF biomarkers and the impairment of different cognitive domains were evaluated as previously described [28]. Lumbar puncture (performed at the L3/L4 or L4/L5 interspace) was used to obtain CSF samples from participants at the Kaohsiung Veterans General Hospital (KVGH) after the participants had provided informed consent. This study was approved by the KVGH Institutional Review Board. Approximately 5 mL of CSF was withdrawn during the lumbar puncture. After collection, samples were spun at 3000 rpm at 4 °C for 10 min to remove any cells or debris and were then transferred to a freezer and stored at −80 °C until use. Only CSF samples without visible blood were centrifuged, and the total Tau protein was further analyzed using Western blot assays in several CSF samples from each of the four groups to ensure that the total protein for each lane loading was 4 μg.

### 2.7. iPSC Lines Genotyping and Exome Sequencing

Normal control iPSC lines, NTUH-iPSC-01-05 and NTUH-iPSC-02-02 (abbreviated as N1 and N2, respectively), were purchased from the Bioresource Collection and Research Center of Food Industry Research and Development Institute, Taiwan. The biopsy samples, one *APP* (^D678H^), one *ApoE4* (^ε4/^^ε4^), and one *PSEN1* (^P117L^), were used to obtain iPSC lines from subjects at the Taipei Veterans General Hospital (TVGH) and upon informed patient consent. Genomic DNA was isolated from peripheral leukocytes using a DNA Extraction Kit (Stratagene, La Jolla, CA, USA). The *APP*, *ApoE4*, and *PSEN1* genotype were determined by PCR amplification, gel purification, HinfI digestion, and direct sequencing using an ABI PRISM 3730 Genetic Analyzer (Applied Biosystems, Forster City, CA, USA). Each PCR included 20 ng of genomic DNA, 0.9 μM of each primer, and Universal PCR Master Mix (Applied Biosystems, Forster City, CA, USA). Such as, direct sequencing of *APP* exon 16 PCR products derived from the patient and from healthy controls revealed a GAC-to-CAC nucleotide substitution in Ab region of the patient’s *APP* gene (in 678th amino acid using *APP*^770^ numbering). The PCR conditions were as follows: 95 °C for 5 min, followed by 40 cycles of 95 °C for 40 s, 58 °C for 30 s, and 72 °C for 40 s, with a final extension at 72 °C for 10 min.

### 2.8. Generation and Culture of Human iPSC

The biopsy samples of AD patients with *APP*^D678H^, *ApoE4*^ε4/^^ε4^, or *PSEN1*^P117L^ mutation were reprogrammed into iPSCs by Vesicular stomatitis virus G (VSV-G) coated Sendai viral transduction of four transcription factors, octamer-binding transcription factor 3/4 (OCT3/4), SRY (sex determining region Y)-box 2 (SOX2), Krüppel-like factor 4 (KLF4), and c-Myc (CytoTune-iPS reprogramming kit, Thermo Fisher Scientific, USA). The establishment of AD-iPSC lines followed the Policy Instructions on the Ethics of Human Embryo and Embryonic Stem Cell Research in Taiwan. In addition, approval from the Ethic Institutional Review Board and informed consent was also obtained from Taipei Veterans General Hospital and National Yang-Ming University. The successful iPSC clones exhibited typical characteristics of pluripotent stem cells and normal karyotypes, and the differentiation ability was confirmed in vivo by teratoma formation assay, and in vitro by the formation of three germ layers via embryoid bodies (Appendix A) as also previously described [29,30]. Human iPSC lines were routinely maintained in Essential 8 Medium (E8, Invitrogen Life Technologies, Carlsbad, CA, USA) on cell culture dishes coated with 0.5 μg/cm^2^ recombinant human vitronectin (Invitrogen Life Technologies, Carlsbad, CA, USA). To passage iPSCs, the cells were washed twice with sterilized Dulbecco’s PBS (DPBS; Invitrogen Life Technologies, Carlsbad, CA, USA) without calcium or magnesium and then incubated with DPBS/ethylenediaminetetraacetic acid (EDTA; 0.5 mM UltraPure EDTA in DPBS) at 37 °C for 3 min. When the cells began to separate and round up, the DPBS/EDTA was removed, and the cells were washed swiftly from the vessel. An appropriate number of cells was transferred to a new culture dish and maintained in an incubator at 37 °C under 5% CO_2_. The cells were subcultured every 4–5 days and then reseeded at a 1:5 to 1:10 ratio. The culture medium was refreshed daily.

#### 2.8.1. Generation of Isogenic iPSC lines Using CRISPR/Cas9 Technology

DNA oligonucleotides used for gRNA targeting were designed with the GeneArt CRISPR gRNA Design Tool (Thermo Fisher Scientific; Invitrogen Life Technologies, Carlsbad, CA, USA). To examine the cleavage efficiency of gRNA, a series of gRNAs flanking the target site was designed and synthesized. Each individual gRNA was combined with Cas9 nuclease (Invitrogen Life Technologies, Carlsbad, CA, USA) to form Cas9 protein/gRNA ribonucleoprotein complexes (Cas9 RNPs). The Cas9 RNPs were then used to transfect the iPSCs, a task for which the Neon Transfection System (Invitrogen Life Technologies, Carlsbad, CA, USA) was used. Genomic editing efficiency was then evaluated through T7 Endonuclease I (T7E1) assay 48 h after transfection. The gRNAs that had both the highest cleavage efficiencies and closest proximity to the target site were selected for the subsequent genome editing. For precise genome editing, the Cas9 RNPs and repair template (ssODN from IDT, Coralville, IA, USA) were coelectroporated into the iPSCs. The transfected iPSCs were then clonally expanded to derive isogenic cell lines. The single-nucleotide substitution was screened using a TaqMan SNP Genotyping Assay (Applied Biosystems, Forster City, CA, USA) and confirmed through Sanger sequencing.

#### 2.8.2. RNA isolation and RT-PCR

Total RNA was isolated using a Tissue Total RNA Mini Kit (Geneaid, Taipei, Taiwan) by following the manufacturer’s instructions. In-column DNase I digestion was performed to remove genomic DNA contamination. Reverse transcription was then performed using Superscript IV (Invitrogen Life Technologies, Carlsbad, CA, USA) and Oligo(dT)_20_ primers by following the manufacturer’s guidelines. For PCR amplification, it was done with a denaturation step at 94 °C for 5 min, followed by 35 cycles of denaturation at 94 °C for 30 s, primer annealing at 60 °C for 30 s, and primer extension at 72 °C for 1 min. Upon completion of the cycling steps, a final extension at 72 °C for 5 min was done and then the reaction was analyzed by DNA gel electrophoresis. The primers for PCR analysis are listed in Appendix A.

#### 2.8.3. Karyotyping

Chromosomal analysis was performed by G-banding analysis at the Cytogenetic Center of Ko’s Obstetrics and Gynecology Clinic, Taipei, Taiwan, following the International System Cytogenetics Nomenclature recommendations.

#### 2.8.4. In Vivo Teratoma Formation

The human induced pluripotent stem cell (hiPSC) colony was dissociated and resuspended in PBS. 2–5 × 10^6^ cells per mouse were subcutaneously implanted into the dorsal flanks of SCID mice. Teratoma growth was monitored weekly and mice were sacrificed at 10 weeks post implantation. Teratomas were collected, fixed in formaldehyde, embedded in paraffin, sectioned and stained with hematoxylin and eosin (H&E) for histological analysis.

#### 2.8.5. In Vitro Embryoid Body (EB) Formation

hiPSC colonies were treated with collagenase IV (GIBCO BRL Life Technologies, Invitrogen Life Technologies, Carlsbad, CA, USA) and gently scraped off the culture dishes. After centrifugation, cells were resuspended in the EB medium (DMEM/F12 supplemented with 20% KnockOut™ Serum Replacement (KOSR, GIBCO BRL Life Technologies, Invitrogen Life Technologies, Carlsbad, CA, USA), 2 mM Glutamax, 1% MEM non-essential amino acids (MEM-NEAA), and 55 µM 2-mercaptoethanol) and transferred into low attachment dishes. EBs were cultured in a 37 °C incubator with humidified atmosphere of 5% CO_2_. The medium was changed every other day. After 14 days EBs were collected and assessed by RT-PCR for expression of stem cell and differentiation markers.

#### 2.8.6. Immunofluorescent Staining

hiPSCs were fixed in 4% paraformaldehyde in PBS for 20 min at room temperature, washed three times with PBS, and then incubated in 0.1% Triton X-100 in PBS for 5 min at room temperature. Next the cells were blocked in PBS containing 3% bovine serum albumin (Sigma-Aldrich, St. Louis, Missouri, USA) and 1% goat serum (GIBCO BRL Life Technologies, Invitrogen Life Technologies, Carlsbad, USA) for 1 h at room temperature. The primary rabbit anti-Oct4 antibody (GeneTex, Irvine, CA, USA) was applied at the indicated concentration overnight at 4 °C. Next the cells were washed in PBS three times and then the secondary antibody, goat anti-rabbit IgG (H+L) – Alexa Fluor 555 (Thermo Fisher Scientific; GIBCO BRL Life Technologies, Invitrogen Life Technologies, Carlsbad, CA, USA), was applied for 1 h at room temperature. Fluorescent images were captured using a Zeiss microscope and an Andor Zyla cMOS camera and then processed using ImageJ software (NIH, Bethesda, MD, USA).

#### 2.8.7. Lentivirus Production and Infection 

The lentiviral vector pTet-O-Ngn2-puro was a gift from Marius Wernig (Addgene plasmid #52047). For lentivirus production, HEK 293T cells were seeded at 5 × 10^6^ cells in a 10 cm dish and incubated overnight. Cells were cotransfected with 6 μg of pTet-O-Ngn2puro or pFUW-rtTA with 5 μg of packaging plasmid pCMV-Δ8.91 and 1 μg of envelope plasmid pVSV-G with Lipofectamine 3000 (Invitrogen Life Technologies, Carlsbad, CA, USA) following the manufacturer’s instructions. The supernatant was collected at 24 and 72 h after transfection, filtered through a 0.45-μm filter to remove cell debris, and then purified using the Lenti-X Maxi purification kit (TaKaRa Bio Inc., Otsu, Shiga, Japan). Virus concentrates were aliquoted at 100 μL and stored at −80 °C until use. For iPSC transduction, approximately 2 × 10^4^ iPSCs in one well of a 24-well plate were infected with lentivirus containing rtTA. The next day, the medium was replaced with fresh E8 with lentivirus containing pTet-O-Ngn2-puro. The lentivirus-infected iPSCs were passaged and expanded for neuronal differentiation 3 days after transduction.

#### 2.8.8. Generation of Induced Neurons from iPSC 

The iPSCs were treated with 0.5 mM EDTA and plated as dissociated cells in E8 medium containing 5 μM Y-27632 (MedChemExpress, Monmouth Junction, NJ, USA) at a density of 10^5^ cells/mL on vitronectin-coated dishes on day −1. On day 0, the culture medium was replaced with N2/DMEM/F12/NEAA containing human brain-derived neurotrophic factor (BDNF, 10 mg/L, Peprotech, Inc., Rocky Hill, NJ, USA), human NT-3 (Neurotrophin-3, 10 mg/L, Peprotech, Inc., Rocky Hill, NJ, USA), mouse laminin (0.2 mg/L, Invitrogen Life Technologies, Carlsbad, CA, USA), and doxycycline (2 mg/L). On day 1, a 24 h puromycin selection (1 mg/L) period began. On day 2, transfected cells were replated in neurobasal medium supplemented with B27/Glutamax (Invitrogen Life Technologies, Carlsbad, CA, USA) containing BDNF and NT-3. On day 5, Ara-C (2 μM, Sigma-Aldrich, St. Louis, Missouri, USA) was added to the medium for 24 h to inhibit proliferation of undifferentiated cells. Subsequently, half the medium in each dish was changed every 3–4 days. Induced neurons were assayed on day 28 in most experiments.

### 2.9. Western Blot Analysis

Protein lysates were prepared as described previously [31]. Western blot analysis was performed with primary antibodies for GFP (sc-9996, 1:2000), Tau 46 (sc-32274, 1:500), and β-actin (sc-47778, 1:1000) were obtained from Santa Cruz Biotech (Santa Cruz, CA, USA).

Phospho-GSK-3α/β (Ser21/9, 9331s, 1:1000), Phosphor-(Ser/Thr) PKA Substrate (9621s, 1:1000), and Phosphor-Tau (Ser396, PHF13, 9632s, 1:1000) were obtained from Cell Signaling. GSK-3β (Clone 7, 610202, 1:1000) and Drp1 (Clone 8/DLP1, 611112, 1:1000) were obtained from BD Transduction Laboratories. Tau 5 (QF215086, 1:1000), PhosphorTau (Thr205, 44-738G, 1:2000), Phosphor-Tau (Thr214, 44-742G, 1:2000), Phosphor-Tau (Thr231, 44-746G, 1:2000) and (Thr262, 44-750G, 1:2000) were obtained from Thermo Scientific. Phosphor-Tau (Ser409, AB9662, 1:2000) was obtained from Merck Millipore. Albumin (ab106582, 1:1000) was obtained from Abcam. HA monoclonal (H9658, 1:1000) was obtained from Sigma-Aldrich. PKA RII Subunits (06-411, 1:1000) was obtained from EMD Millipore.

### 2.10. Statistical Analysis

Statistical significance was evaluated through one-way analysis of variance followed by Tukey’s post hoc test to correct for multiple comparisons. A two-tailed Student’s *t*-test was used to compare data between two groups. Statistical significance was set as *p* < 0.05.

## 3. Results

### 3.1. PKA, GSKIP, GSK3β, and Tau May Form a Local Working Complex

We recently identified residue L130 of GSKIP as a critical point for binding with GSK3β, and the L130P GSKIP mutant resulted in loss of inhibition of neurite outgrowth in human neuroblastoma SH-SY5Y cells [5]. Further studies have demonstrated that mammalian GSKIP favors dimer instead of monomer because the V41/L45 sites are distal to the L130 residue in the GSKIP monomer formation, thus preventing mutual interactions between PKA RII and the GSK3β binding region, indicating that L130 point mutation is essential for the GSK3β binding function [7,9]. To determine whether GSKIP/GSK-3β/Tau could form an assembly, co-IP assay using GFP–Tau was used to pull down GSK3β, GSKIP, and PKA RII, as shown in Figure 1. In addition, the complex binding ability of GSKIP was a total loss in the L130P mutant form (Figure 1, right panel, line 3 in lane 3). Moreover, our data showed that PKA RII could not be pulled down by either V41P/L45P or the L130P mutant (Figure 1, right panel, line 4 in lanes 2 and 3; compare with lane 1), both GSKIP V41P/L45P and GSKIP L130P mutants which resulted in PKA RII disassemble from the Tau/GSK3β complex (Figure 1, left panel, line 4 in lanes 2 and 3; compare with lane 1), indicating complex destruction by the mutants. This is consistent with our previous observations concerning Drp1 and β-catenin [7,9]. Altogether, these data suggest that GSKIP may function as an AKAP and recruit the PKA RII subunits with GSK3β and Tau into close proximity to form a complex (Figure 1).

### 3.2. Knockdown Experiments Revealed that GSKIP and GSK3β Are Involved in cAMP/PKA/Tau Axis Signaling in SH-SY5Y Cells

To protect neurons against oxidative stress, GSKIP L130 and Drp1 phosphorylation are critical points for the interaction between GSKIP/GSK3β and GSK3β/PKA complexes, respectively [7]. Because the Tau protein can interact and form a complex with GSKIP/GSK3β/PKA RII, it is crucial to determine Tau’s role in this complex. Therefore, we experimentally used forskolin (FSK) to activate PKA; the GSKIP V41P/L45P mutant suppressed Tau Ser409 phosphorylation, and GSKIP L130P enhanced Tau Ser409 phosphorylation (Figure 2A). The latter is not expected that may result from versatile conformation of phosphor-Tau. This result suggests that the physical interaction between PKA and GSKIP is related to Tau Ser409 phosphorylation. In addition, after PKA activation, overexpressed kinase-dead GSK3β K85R (retains capacity to bind GSKIP) but not K85M (loss of capacity to bind GSKIP) in SH-SY5Y cells had a higher level of phosphor-Tau at Ser409 (Figure 2B). This result suggests that the physical interaction between GSK3β and GSKIP is related to Tau Ser409 phosphorylation, even under activated PKA. Moreover, in the presence of FSK, the silencing of GSK3β but not GSK3α led to a decrease in Tau Ser409 phosphorylation (Figure 2C). We also examined the phosphorylation level of other sites of Tau in SH-SY5Y cells treated with FSK in different conditions. The results indicate that the phosphorylation levels of Ser409, Ser262, and Ser214 are increased by PKA but that of Tau Ser396 by GSK3β was slightly decreased (Appendix A). The other two sites of phosphorylation of Tau by GSK3β Ser231 and Ser205 may not be involved in the GSKIP mediated pathway under FSK challenge (Appendix A; compare with Figure 2A).

These observations again suggest that GSK3β acts as a scaffold protein that recruits Tau to PKA, probably through Tau phosphorylation at Ser409. These data indicate that GSKIP and GSK3β coordinate to strengthen cAMP/PKA/Tau axis signaling. In addition, we observed that Tau phosphorylation at Ser409 is mediated not through GSK3β activity but through PKA signaling. Notably, this ensures that GSK3β acts solely as an anchor binding protein rather than relying on its kinase activity in this signaling axis.

### 3.3. Site-Specific Effects of Prephosphorylation of Tau by PKA on the Subsequent Phosphorylation by GSK3β and CaMKII

Phosphorylation of Tau at multiple sites is regulated by several kinases, most notably GSK3β (Tau ser205, Tau ser231, Tau ser396; in AD and control brains), PKA (Tau ser214, Tau ser262, Tau ser409; only in AD brains), and CaMKII (Tau Ser212, Tau Ser214, Tau Ser262, Tau Ser356, Tau Ser416; Table 1) [27,32,33,34,35]. To determine the role of Tau Ser409 phosphorylation on the cAMP/PKA/Tau axis and to examine whether the phosphorylation sites are modulated by other kinase phosphorylation priming, we used in vitro kinase assay with phosphorylation-specific Tau antibodies to detect Tau phosphorylation at various sites after incubation of total Tau with PKA, GSK3β, or CaMKII by different order (Figure 3A). The data showed that phosphorylation of Tau at Ser205, Ser231, and Ser396 was associated with GSK3β, whereas phosphorylation at sites Ser214, Ser262, and Ser409 was associated with PKA. Regardless of pre-incubation with PKA, the GSK3β phosphorylation sites on Tau were always the same. However, the PKA phosphorylation sites on Tau exhibited an interesting result in that Tau Ser409 significantly decreased fourfold after incubating with GSK3β (Figure 3A, lanes 4 and 5; compare with lane 3). This result indicated the masking effect of GSK3β, which induced protein conformation alteration that blocked PKA binding with Tau. This effect on Tau phosphorylation was also evident with CaMKII and PKA cotreatment, but it seems to have been unrelated to all PKA Tau sites, including Ser214, Ser262, and even Ser409 (Figure 3B). These results indicate a specific Tau phosphorylation pattern in which each kinase is alone or in any combination of two different kinases, but especially GSK3β plus PKA (Figure 3A). Therefore, we next assessed the hyperphosphorylation of Tau conformation to determine whether this novel finding has added benefit for AD assessments. We undertook experiments to determine the Tau phosphorylation level in the CSF of different types of AD patients to determine whether it could serve as a diagnostic biomarker.

### 3.4. Total and Phosphorylated Tau of CSF in AD, NAD, and MCI Patients

It is well-known that hyperphosphorylation of Tau promotes the accumulation of Tau protein and its binding ability to microtubules [17,35]. Therefore, the Tau phosphorylation level of CSF in AD, NA, and MCI patients may serve as a diagnostic indicator. We first compared CSF Tau and phosphorylation Tau among AD, NAD, and MCI patients. The total Tau concentration, PKA phosphorylation sites on Tau (Ser214, Ser262, and Ser409), and GSK3β phosphorylation sites on Tau (Ser205, Ser231, and Ser396) were evaluated from CSF, as shown in Figure 4. We observed a higher level of phosphor-Tau (Ser214, Ser262, and Ser409) by PKA in the AD group when compared with the normal group, but significant differences were not found among the NAD, MCI, and normal groups. The results also showed that Tau phosphorylation by GSK3β at the Ser205, Ser231, and Ser396 sites occurred in the normal, AD (*APP*^D678H^), NAD, and MCI groups, a finding consistent with those previously reported regarding Tau hyperphosphorylation sites [28,37] that indicated the phosphor-Tau sites in AD and normal brains [38].

### 3.5. Phosphorylation of Tau in AD-iPSC-Derived Neurons with CRISPR/Cas9

Several AD cases were reported to carry the *APP*^D678H^ mutation, which is also called the Taiwanese mutation [24,25,38,39,40,41]. The use of CRISPR/Cas technology for genome editing has many potential applications [26]. To mimic the clinical situation of AD, previously generated iPSCs from an AD patient carrying heterozygous *APP*^WT/D678H^ were further edited using Cas9-gRNA cleavage to create isogenic homozygous control *APP*^WT/WT^ and *APP*^D678H/D678H^ (Table 2, Figure 5A,B; [37]). The isogenic iPSC clones exhibited typical characteristics of pluripotent stem cells: Similar morphology to embryonic stem cells (ESCs) and expression of pluripotent marker OCT3/4 (Appendix A). To extend the clinical significance of Tau phosphorylation status in AD, the isogenic iPSC lines were thus differentiated into gluatamatergic neurons, which are severely lost in AD brains, using the Neurogenin 2 (Ngn2) induction method (Appendix A). The iPSC-derived neurons displayed somatodendritic and axonal markers, microtubule-associated protein 2 (MAP2) and neurofilament (Smi-312), respectively, as well as exhibited neuronal activity (Appendix A). Furthermore, we found no obvious difference in the ability to generate induced neurons between isogenic lines, suggesting the *APP*^D678H^ mutation has no significant effect on neuronal differentiation. To determine whether AD-iPSC-derived neurons recapitulated Tau and phosphorylated AD Tau, we examined the Tau and phosphorylated Tau levels in the AD-iPSC-derived neurons; we found higher levels of PKA phosphorylation on Tau sites Ser214, Ser262, and Ser409 (Figure 5C,D). We also examined the levels of GSK3β phosphorylation on Tau sites Ser205, Ser231, Ser396 and found reduced phosphorylation (Tau Ser231) in the heterozygous *APP*^WT/D678H^ mutation, isogenic control *APP*^WT/WT^, and *APP*^D678H/D678H^ compared with ctrl-iPSCs. This finding indicated that the phosphorylation of Tau Ser214, Ser262, and Ser409 by PKA was increased, whereas the phosphorylation of Tau Ser231 and Ser396 by GSK3β was decreased, resulting in no overall change in total Tau (Figure 5C,D). Further, we examined an intense level of all sites of phosphorylation by PKA (Tau Ser214, Ser262, and Ser409) in heterozygous *APP*^WT/D678H^; the examination revealed elevated expression of GSKIP, suggesting that it is a key player (Figure 5E,F). These results were consistent with a previous study that demonstrated three sites of Tau phosphorylation by PKA, but not by GSK3β, in the AD brain [34]. Furthermore, we did not observe these differences in Tau phosphorylation situations in the groups of *ApoE4* and *PSEN1* patients (Appendix A) [26,42,43].

## 4. Discussion

In this study, we identified a specific PKA phosphorylation site on Tau protein that was accompanied by GSK3β and GSKIP to form a working complex from the neuroblastoma cell line in clinical AD patients. We previously identified residue L130 of GSKIP as being critical for binding with GSK3β, and cells with the L130P GSKIP mutant lose the inhibition of neurite outgrowth in human SH-SY5Y neuroblastoma cells [5]. We also demonstrated that mammalian GSKIP favors dimer instead of monomer formation because the V41/L45 sites are distal to the L130 residue in the GSKIP monomer; this prevents possible mutual interactions between PKA RII and the GSK3β binding region. We also determined that the L130P mutation is crucial for GSK3β binding [7,9]. Our data in Figure 1 reveal that the pET32a-HA-GSKIP (V41/L45P) mutant failed to interact with the PKA RII binding site (loss of function; negative regulation) and increased binding to GSK3β (gain of function; positive regulation), which is consistent with earlier reports [7,9]. These data suggest that GSKIP may function as an AKAP to recruit the GSK3β, Tau, and PKA RII subunits into close proximity and that GSKIP forms a working model with PKA RII/GSK3β/Tau (Figure 6).

Previous studies have revealed that PKA RII directly phosphorylated Drp1 Ser637 via GSK3β as scaffold, and that Drp1 Ser637 phosphorylation was significantly reduced after removing the GSK3β. In addition, the coordination of GSKIP and GSK3β has been shown to enhance the cAMP/PKA/Drp1 axis through the study of GSK3β K85R (retained GSKIP binding ability) and GSK3β K85M (loss GSKIP binding ability) [1,6,7,9]. In this study, both Drp1 Ser637 and Tau Ser409 phosphorylation were enhanced in the GSK3β K85R group and were reduced in the GSK3β K85M group. The phosphorylation signals of Drp1 Ser637 and Tau Ser409 were also reduced after knocking down GSK3α and GSK3β, as depicted in in Figure 2C. According to the results in Figure 2A,B, we can conclude three things. First, the interaction between GSKIP and GSK3β affects the efficiency of the complex, as evidenced by the GSKIP interaction with PKA (V41/L45) and GSK3β (L130). Second, when the cAMP signaling pathway was activated, PKA RII phosphorylated Tau Ser409 with the PKA/GSKIP/GSK3β complex, but GSK3β did not. Third, GSK3β only serves as a scaffold for anchoring PKA and Tau and is not a kinase in the complex.

There are at least 85 putative sites of phosphorylation in Tau, including 80 serine and threonine residues, and nearly half of them are phosphorylated to a certain extent in the AD brain. Tau phosphorylation is regulated by many kinases, including GSK3β, cyclindependent kinase 5, mitogen-activated protein kinases, dual-specificity tyrosine-regulated kinase 1A, PKA, PKB/Akt, PKC, PKN, and CaMKII [27,32,33,34,35,44]. Mixed evidence points to a possible role of phosphor-Tau in AD, especially in the GSK3β character (Tau undergoes prior phosphorylation at a site four amino acids upstream of the phosphorylation site for S/TXXXS/T motif priming or the S/TP motif) [45,46,47]. The precise role of some of these kinases on Tau is still under investigation, with particular interest focused on GSK3β, PKA, and CaMKII. Each of these three kinases or their combinations can phosphorylate Tau at multiple sites (for a summary of the phosphorylation sites see Table 1) [27,30,31,32,33]. When Tau is not interacting with other proteins, it may curl on its own, and this random curled state is thought to be important for preventing interactions with other Tau proteins by masking the possible interacting phosphorylated sites [48,49]. As evident in Figure 3, Tau Ser409, Ser214, and Ser262 were all phosphorylated under a PKA-only condition, whereas Tau Ser409 phosphorylation was dramatically reduced under either both PKA plus GSK3β and GSK3β plus PKA. Hyperphosphorylated Tau is a major constituent of NFTs.

Considerable evidence has suggested that Tau hyperphosphorylation occurs in the early stage of AD and other diseases caused by abnormal Tau expression [50,51,52,53]. The combinations of PKA plus GSK3β that resulted in phosphorylation of Tau site Ser409 by PKA were underestimated (fourfold lower than expected). To determine whether this unique site of Tau Ser409 was identical in clinical application to in different types of AD, we compared Tau phosphorylation in the CSF of normal, AD, MCI, and NAD patients. We determined that the CSF levels of total Tau are elevated in AD compared with the normal control, MCI, and NAD neurological control cases. Similar levels of total Tau were observed among the normal control, MCI, and NAD groups, suggesting that the rate of neurodegeneration in the NAD group was similar to that observed in the non-neurologically aged controls and normal patients (Figure 4A,B). The CSF levels of phosphor-Tau at PKA phosphorylation sites (Ser214, Ser262, and Ser409), especially relative to the total Tau levels, are elevated only in AD (Figure 4A,B). This phosphor-Tau at Ser409 caused by PKA in AD was not compatible with what we observed in in vitro kinase assay (Figure 3A, lanes 4 and 5; compare with Figure 4, lane Ser409 (in AD)). We believe this discrepancy may not be controlled simply by GSK3β or PKA but may be due to some regulatory mechanism at many phosphor-Tau sites, such as protein conformational changes which is versatile and possesses many tasks independently during AD pathogenesis (Figure 6). Tauopathic changes were noted only in the striata of Parkinson disease (PD) and PD with dementia, with increased hyperphosphorylation seen at Ser262 and Ser396 [54].

Tau Ser214 phosphorylation was identified in the early stage of AD and observe to persist through the formation of whole NFTs. Tau phosphorylation at Ser214 also has synergistic actions when combined with other kinases, and this accelerates Tau hyperphosphorylation at multiple sites [55]. Therefore, the specific phosphorylation sites of Tau may yield information concerning AD pathogenesis. These findings are consistent with a previous report in which elevated CSF levels of total Tau were observed in the early stages of AD [56]. Despite the GSKIP regulation of PKA-mediated Tau Ser214, Ser262, and Ser409 phosphorylation, the GSKIP protein concentration in CSF was undetectable in our AD patients, as determined through Western blot (Appendix A). Confirming the interplay between Tau phosphorylation and the PKA/GSKIP/GSK3β complex may require evaluating CSF from *APP* patients. Our findings thus suggest that the increased level of PKA phosphorylation sites (Ser214, Ser262, and Ser409) in CSF is specific to AD and could be used as a diagnostic aid for AD. Furthermore, none of the patients in the control groups had any lesions of abnormally hyperphosphorylated Tau in their brains, and thus they exhibited low CSF levels of abnormally hyperphosphorylated Tau.

The shift of *APP* processing into the amyloidogenic pathway is one of the key factors in AD pathogenesis [57]. We transformed two AD-patient-delivered mononuclear cells to iPSCs, and they differentiated into neurons following the use of CRISPR/Cas9 to create *APP*^WT^ and double ^D678H^ mutants. The *APP*^D678H^ mutant facilitated *APP* sorting into the endosomal–lysosomal pathway and thus increased Aβ production [39]. In this study, the signals of Tau Ser214, Ser262, and Ser409 phosphorylation (PKA target sites; see Table 1) exhibited a significant increase in the neurons derived from the *APP*^WT/D678H^–iPSC group, but the signal in Tau Ser231 (GSK3β target sites) was dramatically decreased. We found that the signals of Tau Ser214, Ser262, and Ser409 phosphorylation (PKA target sites) were reduced after editing back to ^WT/WT^ (Figure 5B,C). In addition to serving as the GSK3β blockade substrate, GSKIP also inhibited GSK3β onto phosphor-Tau through negative regulation. By contrast, a recent report showed that the neurons derived from *APP*^D678H^– iPSC exhibited aberrant accumulation of intracellular material and secreted Aβ42 and Aβ40 in GSK3β hyperphosphorylation of Thr181 and Ser396 [25]. We believe the lack of change in the GSKIP protein level in the *APP*^D678H/D678H^–iPSC homologous mutant could be due to two possible mechanisms: The homo-type of *APP*^D678H/D678H^ may cause serious damage to iPSCs, or CRISPR/Cas9 technology may cause an off-target effect.

Since the continuous failure of Aβ-targeted therapeutics, the Phosphor-Tau has received much attention and highlights the urgency to consider alternative therapeutic strategies for AD [18]. In conclusion, we suggest that the PKA/GSKIP/GSK3β/Tau complex also plays a key role on the development of AD. In addition, total Tau and PKA phosphor Tau sites (Ser214, Ser262, and Ser409) in CSF and iPSCs are good feasibility of standard criteria for the detection of AD. Altogether, this study provides compelling evidence of the function of both GSKIP and GSK3β as anchoring proteins that enhance cAMP/PKA/Tau axis signaling during AD pathogenesis (Figure 6). Our findings suggest that phospho-Tau resulting from PKA and GSKIP could serve as a potential biomarker and could represent a new strategy for the detection and determination of the mechanism of pathogenesis in AD. Thus, integrating cAMP/PKA/Tau and GSKIP and Aβ42 and Aβ40 biomarkers for the detection of different types of AD and tauopathies and for the prediction of treatment outcomes could improve the clinical practice of precision medicine.

## Figures and Tables

**Figure 1 jcm-08-01751-f001:**
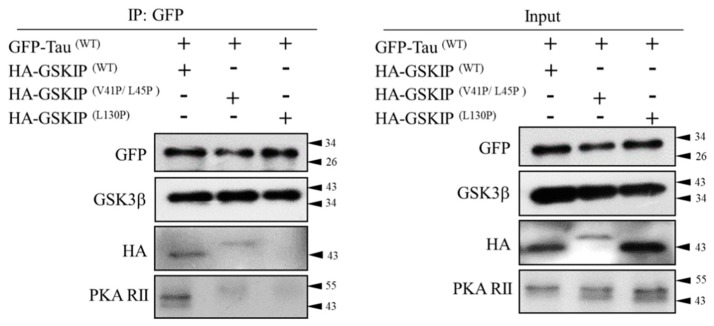
Tau interacts with glycogen synthase kinase 3β (GSK3β), GSK3β interaction protein (GSKIP), and protein kinase A (PKA) in HEK293 cells. pEGFP-C1-Tau, pET32a-HA-GSKIP (L130P), or pET32a-HA-GSKIP (V41/L45P) transfected cells were collected, and total lysates were subjected to IP using anti-GFP antibody. The resulting precipitates were then analyzed through immunoblotting with anti-GFP, GSK3β, HA, and PKA antibodies.

**Figure 2 jcm-08-01751-f002:**
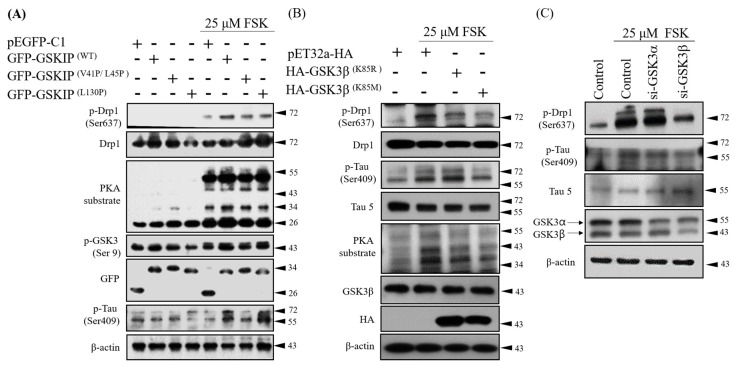
GSKIP through GSK3β-mediated anchoring to modulate Tau phosphorylation by PKA kinase. (**A**) pEGFP-C1 or pEGFP-C1–GSKIP wt and mutants were transiently expressed in SH-SY5Y cells. The cells were untreated (left four lanes) or incubated with 25 μM forskolin (FSK) for 1 h (right four lanes). (**B**) Kinase-inactive pET32a-HA-GSK3β-K85R or -K85M mutants were transiently expressed in SH-SY5Y cells. pET32a-HA-GSK3β-K85R mutant retained the capacity to bind GSKIP, but such capacity was not evident in pET32a-HA-GSK3β-K85M. Cells were incubated with 25 μM FSK for 2 h and were then analyzed through Western blotting. (**C**) Knockdown of GSK3β but not GSK3α expression altered phosphorylation of dynamin related protein 1 (Drp1) Ser637 and Tau Ser409.

**Figure 3 jcm-08-01751-f003:**
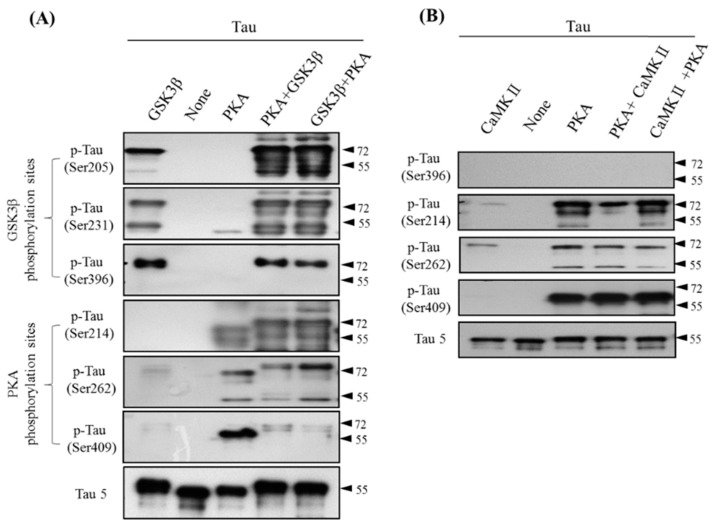
Site-specific phosphorylation of Tau by PKA or GSK3β and Ca2+/calmodulin-dependent protein kinase II (CaMKII). (**A**) Site-specific effects of prephosphorylation of Tau by PKA on the subsequent phosphorylation by GSK-3β. Recombinant Tau was phosphorylated by GSK3β and PKA for 3 h (lines 1, 3, respectively); line 2 shows the control-treated Tau. Tau was first phosphorylated by PKA for 1 h and then incubated with GSK3β kinase for 120 min (line 4); then, it was phosphorylated by GSK3β for 1 h, followed by incubation with PKA kinase for 120 min (line 5). (**B**) Site-specific effects of prephosphorylation of Tau by PKA on the subsequent phosphorylation by CaMKII.

**Figure 4 jcm-08-01751-f004:**
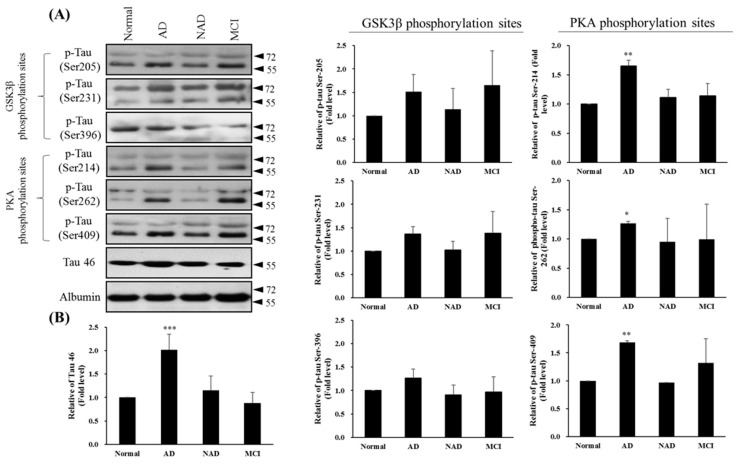
Western blot of Tau46 (total Tau) and phosphorylated Tau by GSK3β and PKA kinases from cerebrospinal fluid (CSF) sampling of four groups: normal, Alzheimer’s disease (AD), neurological disorders (NAD), and mild cognitive impairment (MCI). (**A**) Western blotting with Tau 46, Tau Ser205, Tau Ser231, and Tau Ser396 by GSK3β or Tau Ser214, Tau Ser262, and Tau Ser409 by PKA antibodies. Albumin was used as an internal control. (**B**) Statistical analysis. Bar graphs represent the mean ± SD of triplicates. * *p* < 0.05, ** *p* < 0.01, *** *p* < 0.001 compared with the control group.

**Figure 5 jcm-08-01751-f005:**
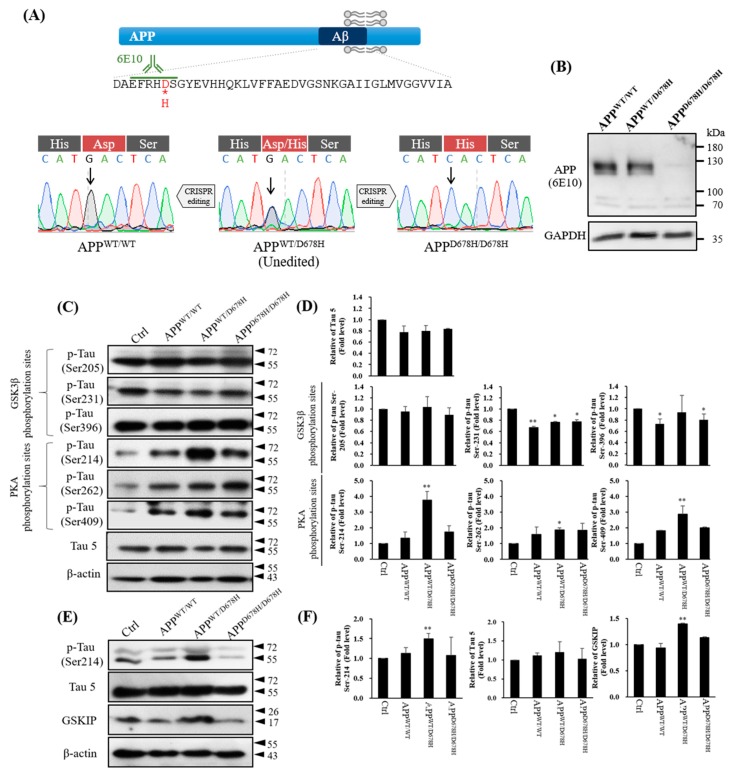
Tau phosphorylation in a patient-derived Alzheimer’s disease induced pluripotent stem cell (AD-iPSC) line with amyloid precursor protein (*APP*^WT/D678H^) mutation cultured at 29 days. Isogenic control (^WT/WT^) and isogenic mutant (^D678H/D678H^) were generated by CRISPR/Cas9 gene editing and examined in parallel. (**A**) The diagram presents *APP* protein and the ^D678H^ mutation within amyloid-β (Aβ) region in AD-iPSC (upper panel). Direct sequencing of *APP* exon 16 PCR products derived from the patient and from healthy controls revealed a GAC-to-CAC nucleotide substitution in Aβ region of the patient’s *APP* gene (in 678th amino acid using *APP*^770^ numbering or in 7th amino acid using Aβ numbering) (lower panel). (**B**) Western blotting with 6E10 antibody, which recognizes residues 3-8 of Aβ, was used to detect the expression of *APP* protein in D29 iPSC-derived neurons. GAPDH was used to confirm similar protein loading across samples. (**C**) Western blotting with antibodies of Tau 5, Tau Ser409, Tau Ser231, Tau Ser396 by GSK3β or Tau Ser214, Tau Ser262 and Tau Ser409 by PKA. Tau 5 (Total Tau) was used as an internal control compared with the control group. β-actin was used as an internal control. (**D**) Statistic analysis for (C). (**E**) Western blotting with GSKIP antibody. (**F**) Statistic analysis for (E). Bar graphs represent the mean of triplicates ± SD. * *p* < 0.05, ** *p* < 0.01, *** *p* < 0.001.

**Figure 6 jcm-08-01751-f006:**
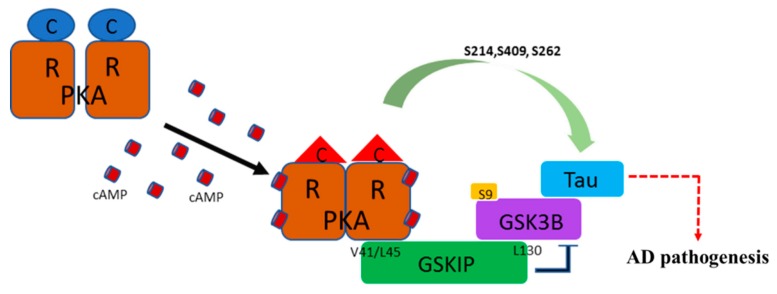
Schematic of GSKIP, GSK3β, PKA RII, and Tau complex. While activating cAMP signaling, GSKIP wt conjugated with activated PKA to facilitate GSK3β Ser9 phosphorylation. GSKIP negatively regulates GSK3β activity (previously characterized in [1,6,7,9]); resulting in double inhibition of GSK3β activity and mediation of Tau phosphorylation by active catalytic form of PKA (red triangle). GSKIP anchors Tau through L130 by tethering GSK3β for PKA-mediated phosphorylation of Ser214, Ser262, and Ser409. This is thought to cause neuron tangles through Tau hyperphosphorylation. R, regulatory subunit of PKA; C, catalytic subunit of PKA.

**Table 1 jcm-08-01751-t001:** List of Tau-specific phosphorylation sites for PKA, GSK3β, and CaMKII.

Kinase	Specific Phosphorylated Site for Tau	Ref.
PKA	Ser214 Ser262 Ser409	Liu et al., 2006 [35], Liu et al., 2008 [32], Zhu et al., 2010 [33], Oliveira et al., 2017 [34]
GSK3β	Ser181 Ser205 Ser231 Ser396 Ser400 Ser404	Liu et al., 2006 [35], Wang et al., 2007 [36], Liu et al., 2008 [32], Oliveira et al., 2017 [34]
CaMKⅡ	Ser212 Ser214 Ser262 Ser356 Ser416	Liu et al., 2006 [35], Wang et al., 2007 [36], Seward et al., 2013 [27], Oliveira et al., 2017 [34]

PKA, protein kinase A; GSK3β, glycogen synthase kinase 3β; CaMKII, Ca2+/calmodulin-dependent protein kinase II.

**Table 2 jcm-08-01751-t002:** Human iPSC (hiPSC) lines.

hiPSC Lines	Gender	Age at Biopsy	Genotype *	APOE Status	Clinical Diagnosis at Time of Biopsy	Note
Ctrl	Female	45	WT	ε4/ε4	N/A	Unedited
APP^WT/WT^	Female	N/A	APP^WT/WT^	ε3/ε4	N/A	CRISPR-edited
APP^WT/D678H^	Female	63	APP^WT/D678H^	ε3/ε4	progressive memory and behavior problems	Unedited
APP^D678H/D678H^	Female	N/A	APP^D678H/D678H^	ε3/ε4	N/A	CRISPR-edited

* Exome sequencing of *APP*, PS1 and PS2 genes.

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
