# Peer review of "GSKIP-Mediated Anchoring Increases Phosphorylation of Tau by PKA but Not by GSK3beta via cAMP/PKA/GSKIP/GSK3/Tau Axis Signaling in Cerebrospinal Fluid and iPS Cells in Alzheimer Disease"

_jcm, 2019, doi:10.3390/jcm8101751_

Round 1

Reviewer 1 Report

The authors have adequately addressed all the concerns I had in my previous review.

Author Response

Response to Reviewer 1:

Thank you for accepting our manuscript. We have cited one more recent paper to the introduction section shown on page 2 (lines 87-89).

Reviewer 2 Report

The manuscript entitled “GSKIP-mediated anchoring increases phosphorylation of Tau by PKA but not by GSK3beta via cAMP/PKA/GSKIP/GSK/Tau axis signaling in cerebrospinal fluid and iPS cells in Alzheimer disease” done by Huey-Jiun et al., conducted in vitro assays of various kinase combinations to show that a combination of GSK3β with PKA which confirmation to the phosphorylation of Tau Ser409.

Exploring their studies on beneficial characteristics of the evaluation of cerebrospinal fluid (CSF) to extend the clinical significance of Tau phosphorylation status in Alzheimer disease (AD), neurological disorders (NAD) and mild cognitive impairment (MCI) are potentially validated and high significance.

Moreover, author used the CRISPR/Cas9 system to produce APPWT/D678H isogenic mutants, activities are potentially sound of this manuscript and in vitro biological validations are interesting and more valuable.

The author explore results the phosphor-Tau caused by PKA, but not GSK3β, in the CSF and iPSCs of AD, and GSKIP and GSK3β function as anchoring proteins to strengthen the cAMP/PKA/Tau axis signaling during AD pathogenesis.

The methodology followed was really impressive and interesting. The authors have well utilized the in vitro biological validations and clinical sample testing procedure more authenticated data in validation of experiments and validations to characterize the kinase specific phosphorylation to validate the PKA phosphorylated Tau sites more valuable to determination of the mechanism of pathogenesis in AD. The methods were explained in a detailed and systematic manner and the corresponding results were discussed in an interactive way. The paper has been written well with clear conclusions. Beyond the potentiality of the manuscript, I have a few minor concerns that the authors may address.

Minor: Consider the more recent studies in phosphorylated Tau studies like Cells 2019, 8(3), 260; https://doi.org/10.3390/cells8030260 are more added value to this manuscript.

Author Response

Response to Reviewer 2:

Thanks for your suggestion. We have added some words in introduction section shown on page 2 (lines 87-89) and cited the paper you suggested as reference 18.

This manuscript is a resubmission of an earlier submission. The following is a list of the peer review reports and author responses from that submission.

Round 1

Reviewer 1 Report

This manuscript describes tau phosphorylation mechanism mediated GSKIP. Tau phosphorylation is a key pathological event initiating tau aggregation process. Therefore, It is important to identify the regulating mechanism.  In this manuscript, the authors  claim that tau phosphorylation occurs by PKA, not by GSK3b, and  GSK3b plays a role in anchoring to tau to PKA/GSKIP complex. Their claims are novel and extremely interesting in the field.  However, the data shown here are insufficient for supporting their claims and most of the data lacks control experiments. Therefore, this manuscript is not suitable for any publication. 

[Figure1] 

1. GFP-Tau only control is required for the evaluation of the interaction between tau and GSK3b in the absence of GSKIP. 

2. HA-GSKIP(V41P/L45P) expression level is comparably low. Repeat experiment and quantification analysis is required to evaluate the significance of the interaction.  

[Figure 2]

1.(2a)Upon the activation of PKA by Forskolin, TauS409 phosphorylation occurs regardless of  GSK3b recruitment by GSKIP.  The result directly suggests that GSK3b anchoring event is not critical for tau phosphorylation. Any explanation?

2 (2b) There is no over-expression control of GSK3b-wt to evaluate the mutant effects.  It is not fair to compare the effects of GSK3b mutants with the endogenous GSK3b.  

3.  Figure 2C is conflicted with Figure 2A in the role of GSK3b. 

[Figure 3] 

1. Simply it is impossible to understand the results and the meaning.  In their experiment, PKA alone is enough to phosphorylate tau regardless of the presence of GSKIP/GSK3b complex formation.  GSK3b strongly phosphorylates tau competing with PKA. Any explanation?   As PKA does, CaMKII also phosphorylates tau409 residue, but the co-treatment reduced 409 phosphorylation. Any explanation?  

[Figure 5] 

1. The authors did not provide any single characterization data of APPWT/D676H (Patient AD-iPSC) cell line and also genome edited CRISPR/Cas9 wild type control and isogenic homozygous control.  It is difficult to evaluate the effects of cell types only with tau phosphorylation levels.  

Reviewer 2 Report

The authors have detailed one of the potential mechanisms of tau hyperphosphorylation implicated in AD pathology via the cAMP/PKA/GSKIP/GSK3/Tau axis signaling. The manuscript is well-written, and the results have been described adequately. The conclusions are corroborated by relevant results. However, some improvements that will further improve the quality of the manuscript are suggested as follows –

1.     In figure 1, the authors describe the right panel in detail, however, the left panel has not been elaborated. The reviewer suggests that authors should elaborate the left panel.

2.     The authors claim that PKA RII could not be pulled down by either V41P/L45P or the L130P mutant 278 (Figure 1, right panel, line 4 in lanes 2 and 3; compare with lane 1), however, a diffuse band is observed in all three lanes in line 4 of right panel. The authors should explain this discrepancy.

3.     The loading control is missing for Figure 2B.

4.     In figure 3 and 5, Tau 5 antibody has been used, however, Tau 46 was the choice of the antibody in analysis of total tau from CSF. Was there as specific reason for choosing two difference anti-tau antibodies?

Overall, the manuscript meets the quality for publication in JCM and should be accepted once these minor revisions have been addressed.
